# Does the Digital Economy Increase Green TFP in Cities?

**DOI:** 10.3390/ijerph20021442

**Published:** 2023-01-12

**Authors:** Chuanyu Zhao, Zhongquan Liu, Xianfeng Yan

**Affiliations:** 1Business School, Ningbo University, Nningbo 315211, China; 2Center for Innovation-Driven Development, National Development and Reform Commission, People’s Republic of China, Beijing 100038, China; 3Center for Digital Economy Research and Development, National Development and Reform Commission, People’s Republic of China, Beijing 100038, China; 4School of Management, Taizhou Vocational and Technical College, Taizhou 318000, China

**Keywords:** digital economy, green productivity, green patents

## Abstract

COVID-19 accelerated the growth of the digital economy and digital transformation across the globe. Meanwhile, it also created a higher demand for productivity in the real economy. Hence, the correlation between the digital economy and green productivity is worth studying as COVID-19 prevention becomes the norm. The digital economy overcomes the limitations imposed by traditional factors of production on economic growth and empowers innovative R&D and resource allocation in all aspects. This study delved into the digital economy by focusing on its green value at different levels of development. The study gathered the green-productivity indices and the principal components of the digital economy for each prefecture-level city in China from 2011 to 2019 and meticulously portrayed their trends in spatial and temporal figures. Meanwhile, regression models were used to verify the mechanism through which digital-economy development influences the changes in green productivity. The results showed that: (1) a higher level of digital economy helps to increase urban green total-factor productivity (GTFP) and that the conclusions of this paper still held after potential endogeneity problems were solved through the instrumental-variables approach; (2) the digital economy will drive an increase in urban GTFP by upgrading firms’ production technologies and that digital-economy development encourages green patent applications from firms; and (3) as the digital economy develops, it will also drive urban GTFP increases by removing polluting enterprises from the market and that the higher the level of digital-economy development, the greater the number and probability of polluting enterprises exiting the market. In view of this study’s results, the government should increase the importance of the digital economy, strengthen the role of the digital economy in promoting urban green development, and provide more guidance on regional green development with the help of the digital economy.

## 1. Introduction

While China’s economic development records remarkable achievements, it also faces pressure from both serious resource depletion and environmental pollution. With the strong emphasis on green and sustainable development, the diminishing marginal returns of traditional factors of production and rapidly growing industrial added value are entering into increasing tension with the continuously increasing emission and discharge of pollutants [1]. Therefore, it is necessary to find a development path that balances productivity and green sustainability. Traditional total-factor productivity (TFP) is measured based on the influence of capital and labor input on output without taking into account resource input and the impact on the environment [2]. Therefore, traditional total-factor productivity cannot accurately reflect changes in socioeconomic welfare, which are crucial for industrial policymaking, and it is necessary to conduct in-depth research on the main factors driving green total-factor productivity.

The digital economy is a new economic form that relies on information technology to drive economic growth, and the rapid development of information technology is an important element of the digital economy. Digital technology allows business entities to cut down on costs and increase productivity [3,4,5]. Some scholars believe that the digital economy itself is a very special economic form in which transactions for goods and services are completed virtually. Thus, the digital economy develops in line with the development of information and communication technology (ICT) and relies on the rapid development of information technology to penetrate into all aspects of life, change their respective modes of operation, and improve efficiency [6,7,8]. The core elements of the digital economy include ICT and digital-technology development. Digital technology combined with the manufacturing and service sectors can transform the traditional production process by upgrading manufacturing and servicing processes. Digital technology plus manufacturing is the main trend in the future development of the digital economy, and the development of the digital economy will spawn new business models such as the platform economy and the sharing economy with certain green features. The innovative features of the digital economy will accelerate the formation of the platform economy, and digital platforms will facilitate the search for product and service information, reduce the cost of product and service matching, and increase transaction speeds. The major difference between the digital economy and the traditional economy is in the ways in which the two information flows are presented, with the former based on physical methods and the latter mainly occurring through digital flows. The digital economy can be divided into three scopes based on its fields of application: core, narrow scope, and wide scope, which cover various industries in society. A new production factor—data—overcomes the limitations associated with traditional industry such as high pollution, high inputs, and low outputs while optimizing the production process. At the same time, it is believed that the integration of the digital economy and real manufacturing is the future development trend and that the digital economy will give rise to a series of new business models such as live-streaming-influencer businesses and bicycle sharing. These business models have certain green features and can increase urban GTFP to a certain extent. The innovative features of the digital economy will accelerate the formation of the platform economy. Digital platforms enable quick searches for and the matching of product and service information, as well as improvements in transaction efficiency. In addition, they have energy-saving and emission-reduction characteristics, which can increase the development of GTFP [9,10].

The study of the relationship between the digital economy and total-factor productivity is currently a major concern of national policymakers. Some studies have shown that the development of the digital economy is related to improvements in total-factor productivity. From a macro perspective, digital-economy development can improve regional total-factor productivity as based on a quasi-experiment on national big-data comprehensive pilot zones, and digital technology will lead to improvements in total-factor productivity through the spatial spillover effect [11,12]. From a micro perspective, digital transformation can improve the total-factor productivity of enterprises because digital technology can be used to search for product and service information more efficiently, reduce product and service matching costs, and help transactions to be concluded as soon as possible, thereby improving the production efficiency of enterprises [13,14].

However, these studies have not yet delved into the impact of the digital economy on green total-factor productivity. Since the digital economy can play an important and positive role in the green low-carbon and sharing economy, how the digital economy encourages green development in the context of resource depletion and environmental pollution is a key issue worthy of research. At present, studies related to research on the impact of the digital economy on green development are relatively rare, which underlines the necessity of studying the effect of digital technology on green development. The research on the digital economy focuses on the transformation and upgrading of manufacturing by the digital economy, and there has been no in-depth discussion on the innovative characteristics and green value of such a new production factor as the digital economy. In addition, the digital economy is an inclusive concept with various types, and previous studies were mostly centered on R&D and product sales; little attention was paid to the study of the effect of the digital economy on economic efficiency, and less research was conducted related to the study of the effect of the digital economy on the increase in productivity in the real economy. More importantly, the innovative characteristics of the digital economy are by no means simple network behaviors, but their combination with traditional factors of production generates new factors and creates strong innovation drivers to achieve value creation [15,16,17]. There is a lack of academic studies on the relevant influencing factors and mechanisms. In addition, although most studies affirmed the positive impact of the digital economy on total-factor productivity, they failed to fully reflect the efficiency advantages in terms of improving ecology, the environment, and resource conservation, and generally ignored its relationship with the green development of total-factor productivity. In the face of the multiple dilemmas posed by increasing resource constraints and environmental pressures, enhancing green total-factor productivity will be the key to driving green and sustainable development in the future. The fundamental issue that needs to be addressed for green TFP growth is the shift from factor-driven to technology- and innovation-driven growth in order to maximize output in an ecologically and environmentally friendly manner.

Regarding the connection between the digital economy and green productivity, Lyu et al. (2022) and Liu et al. (2022), for example, argued that the digital economy can significantly improve China’s GTFP. The higher a city’s GTFP, the greater the increase in urban GTFP as a result of the digital economy. Moreover, it was found that the digital economy improved urban GTFP by upgrading industrial infrastructure and alleviating factor-market distortion [18,19]. Indeed, the existing literature offers a beneficial exploration of the connection between the development of digital economy and green productivity, although it mainly discusses the impact of the digital economy on GTFP at the macro level. Enterprises are the main bodies of digital transformation; green productivity requires enterprises to apply green technology, but it also depends on the elimination of polluting enterprises from the market. Therefore, previous research needs to be supplemented by a greater focus on the micro level.

## 2. Channel of Influence

As a key production mode based on the use of digital knowledge and information, the digital economy overcomes the limitations imposed by traditional production factors on economic growth, and it empowers the process of innovation and resource allocation in all aspects. As a low-carbon green technology and production mode, the digital economy not only contributes to the improvement of ecology and the environment in economic development, but also makes it possible to increase green total-factor productivity [20,21,22,23]. Moreover, the efficient transmission efficiency of the digital economy makes it easier to share information between regions, while the spatial externalities created by its technology spillover draw the economic activities between different regions closer together, which may have direct or indirect effects on the green total-factor productivity of other regions.

With the vigorous development of the digital economy, modes of production and lifestyles are undergoing rapid changes. Industrial digitization and digital industrialization are increasing, and people’s lives are becoming increasingly networked and intelligent. The continuous emergence of new technologies and models increases the updating and iteration of information-industry knowledge and technology, while the life cycle of products or services is gradually shortened. Production and business innovation require information enterprises to obtain new market and technical information in time. Convenient and rapid access to technology spillovers is of great value to enterprises. Digital technology effectively reduces the communication costs between enterprises and between regions, unblocks the channels of knowledge spillovers, builds a tight knowledge-flow network, and enables enterprises to leverage technology spillovers more effectively. Enterprises in a given region can freely share innovative resources through this network; digital technology plays an important role in this network by increasing the spread of green innovative technology and other resources. It can not only facilitate the regional exchange of innovative resources between enterprises, but also increase the inter-regional flow of innovation factors and resources and optimize their allocation in the whole economic system.

Industrial digital transformation has also greatly accelerated the informatization process of enterprises. Through the application of new technologies such as big data, cloud computing, blockchain, and the Internet of Things, traditional manufacturing enterprises can become intelligent manufacturers, thereby improving the technological innovation ability of enterprises and thus improving the total-factor productivity of enterprises. From the perspective of entrepreneurship, the capital-entry threshold of the information-service industry itself is relatively low. As high-speed-rail cities can maintain closer communication and technical contact with central cities and developed areas, professionals with technical skills are more likely to enter high-speed-rail cities to start businesses in the information-service industry. In particular, this applies to the professional and technical personnel in central cities or developed areas who are interested in starting businesses in areas where the information industry is still underdeveloped.

Competition is the essential feature of the development of the digital economy. Network effects strengthen both monopolies and competition. Evans and Schmalensee (2007) pointed out that through indirect network effects, incumbents—with their first-mover advantage—generate positive-feedback effects with users flocking to a few platforms, thereby forming an oligopoly trend [24]. However, the existence of the positive-feedback effect would not lead to monopolies in most multilateral markets. After analyzing the search-engine market specifically, although the current market is structured as an oligopoly, the main body of competition is still diversified.

Therefore, this paper proposes two ways in which the digital economy could influence urban GTFP:

First, digital technology encourages resource exchanges between innovative entities. It breaks the barriers of time and space and broadens the channels and scope of information dissemination. A large amount of information can be rapidly stored and shared by innovative entities through cooperation, which increases efficiency and lowers the cost of sharing and acquiring knowledge. Companies in a region can share innovative resources at will through the network. Digital technology contributes to this by facilitating the transmission of such resources as green innovative technologies. Beyond its values in encouraging the regional exchange of innovative factors resources and resources between companies, it also accelerates their flow across regions and optimizes their allocations within the whole economy.

Second, the development of the digital economy can reduce search, transaction, matching, and replication costs effectively by alleviating information asymmetry, thus lowering transaction barriers, breaking market boundaries, expanding market scope, facilitating the flow of factors in a larger space, and optimizing factor allocation. The development of the digital economy rules out traditional pollution- and energy-intensive industries and improves GTFP.

This paper offers the following academic contributions. First, the existing literature focuses mostly on the impact of the digital economy on the traditional industrial-sales model and rarely on production. This paper examines the impact of the digital economy on total-factor productivity in terms of its green-development characteristics, which is a much-needed addition to the existing literature. Second, in order to more accurately measure total-factor productivity in China, the traditional total-factor productivity indicators need to be transformed and upgraded to include factors such as environmental pollution and resource consumption. Therefore, this paper used the DEA–SBM model to combine traditional total-factor productivity with relevant non-desired outputs and used the GML productivity index to measure the green total-factor productivity by region.

## 3. Methodology and Data

### 3.1. Econometric Model

According to the theoretical mechanism described in the last section, the following regression model was established to analyze the impact of the digital economy on green total-factor productivity:(1)$$GTFP_{it}=\beta_{0}+\beta_{1}Digit_{it}+\beta_{2}Z_{it}+\mu_{it}+v_{it}+\epsilon_{it}$$
where i and t represent cities and years, respectively; $GTFP_{it}$ denotes green total-factor productivity; $Digit_{it}$ denotes the level of digital-economy development of City i in Year t; $Z_{it}$ denotes the controlled variables used in this paper such as population, share of secondary industry, share of tertiary industry, share of fixed-asset investment, share of real-estate investment, share of local fiscal expenditure, share of local fiscal revenue, and share of foreign direct investment; $\mu_{it}\;\mathrm{and}\;v_{it}$ denote the city and time fixed effects, respectively; and $\epsilon_{it}$ denotes the potential random error. While considering possible variable omission and the cause–effect relationship, the following verification will address potential endogeneity problems through an instrumental-variables approach.

### 3.2. Definition and Measurement of the Digital Economy

Currently, there is no international standard for the selection of indicators and measurement methods for the digital economy, and there are no unified measurement indexes of the digital economy as a normative guide. For example, during the 5th IMF Statistical Forum, which was themed “Measuring the Digital Economy”, it was mentioned that there was no statistical way to measure the marginal contribution of digital economy to manufacturing products and services. The Digital Economy Competitiveness Index released by the Shanghai Academy of Social Sciences analyzes the development of the digital economy in the world from four aspects through the construction of an international competitiveness model: infrastructure development, industry volume, innovation capacity, and governance evaluation related to digital industries. The Digital Economy Board of Advisors (DEBA) of the U.S. Department of Commerce, the Organization for Economic Cooperation and Development (OECD), the Bureau of Economic Analysis (BEA) of the U.S. Department of Commerce, the European Union (EU), and the China Academy of Information and Communications Technology (CAIC) have all conducted in-depth studies on the measurement methods of the digital economy, but no method has been universally agreed upon; the digital-economy-related measurement method proposed by each international organization has certain limitations on its applicability.

This paper draws on the method of Huang Huiqun et al. (2019), which uses indicators concerning the Internet-access rate, related employee profiles, related output profiles, and cell-phone penetration rate; to be specific, these respectively entail the number of users that have access to broadband Internet per 100 people, the ratio of employees in the computer service and software industry to the total employee population in urban areas, the total amount of telecommunication services per capita, and the number of cell-phone users per 100 people [25]. The original data for these indicators can be obtained from the *China City Statistical Yearbook*. For the development of digital finance, the China Digital Inclusive Finance Index was used; it was jointly compiled by the Peking University Digital Inclusive Financial Index (pku.edu.cn, accessed on 1 October 2022.) and Ant Financial Services Group (Guo Feng et al. 2020) [26]. The comprehensive digital-economy-development index was then obtained by standardizing the data for the five indicators above and then breaking them down through a principal component analysis.

### 3.3. Concept and Measurement of Green Total-Factor Productivity

Green total-factor productivity (GTPF) is measured in a way that incorporates resource and environmental factors into the framework of productivity analysis, which is in line with the concept of green development in the new era. In measuring green total-factor productivity, Chung et al. (1997) first proposed the directional distance function, based on which Malmquist–Luenberger (ML) productivity is measured; this index can measure the pollutant output in the production process and incorporate it into the total-factor-productivity index system [27]. Regarding the measurement of green total-factor productivity, Chung et al. (1997) were the first to introduce pollution emissions to the total-factor-productivity measurement framework based on the directional distance function (DDF) and ML index. Tone (2001) made a related improvement by establishing a slacks-based measure of efficiency based on a directional distance function (SBM-DDF), which effectively reduced measurement bias. Yuan et al. (2015), on the other hand, proposed another measurement of green total-factor productivity with a dynamic time-series effect based on the SBM-DDF function [28,29]. In order to explore the influencing factors of green total-factor productivity in depth, the existing literature focuses on the role of environmental regulation, FDI, technological progress, and carbon emissions in green total-factor productivity. These measurements have three shortcomings: first, there are difficulties in incorporating resource consumption and environmental pollution variables into the specific production function; second, they cannot reflect the directionality of non-desired and desired outputs; third, even though the second shortcoming is solved by model changes using the directional-distance function proposed by Chung, there are strict requirements for radiality and angularity, which further restrict its application range [1,30,31,32].

This paper measured urban total-factor productivity growth through a global data envelopment analysis (DEA) that integrated the super-efficient SBM model while considering the non-expected output and the Malmquist productivity index. The global DEA used the input–output data of all decision makers over the whole period to construct the optimal production frontier and measured all decision makers in different periods within the global optimal production frontier, which effectively solved the problems of infeasible solutions and incomparability across periods.

Inefficiency obtained from the traditional DEA model is subject to the influences of the external environment, random interference, and inefficient management; the traditional DEA cannot overcome their influences. Therefore, a second model similar to the SFA model was established based on the traditional DEA model:(2)$$s_{ik}=f^{i}\left(z_{k};\beta^{i}\right)+v_{ik}+\mu_{ik}$$
where $s_{ik}$ is the slack variable input of item *i* in the *k*-th decision-making unit, $z_{k}$ is the external environment, and $\beta^{i}$ is the index estimated (generally expressed as $f^{i}\left(z_{k};\beta^{i}\right)=z_{k}\beta^{i}$). The error of this model is the mixed standard error ($v_{ik}+\mu_{ik}$), which satisfies $v_{ik}\sim N\left(0,\sigma_{vi}^{2}\right)$ and $u_{ik}\sim N^{+}\left(0,\sigma_{ui}^{2}\right)$, where $v_{ik}$ and $u_{ik}$ are independent of each other and $z_{k}$. $\gamma =\sigma_{ui}^{2}/\left(\sigma_{vi}^{2}+\sigma_{ui}^{2}\right)$ yields the proportion of technical-inefficiency variance in the total variance. When γ is close to 1, the management factor takes the dominant role; when γ is close to 0, the random error takes the dominant role. Next, the SFA’s regression results were used to adjust the inputs by increasing external environment to put DUM in the same context, thus removing the influence of environmental or random factors:(3)$$\hat{x_{\mathrm{ik}}}=x_{ik}+\left[\max\left\{z_{k}\beta^{i}\right\}-z_{k}\beta^{i}\right]+\max\left[\left\{\hat{v_{ik}}\right\}-v_{ik}\right]$$
where $x_{ik}$ denotes the original input, $\hat{x_{\mathrm{ik}}}$ denotes the adjusted value, $\beta^{i}$ is the estimated index of the environmental variable, and $v_{ik}$ is the estimation of the random interference. A third-stage DEA analysis with an adjusted input and original output yields the efficiency without the impact of environmental factors and random interference.

### 3.4. Enterprise Green Innovation

In this study, the green innovation of enterprises refers to innovations in green energy, green production, and green products. Green energy refers to the technical innovation of using renewable energy sources such as solar energy and new materials; green production refers to the technical innovation of improving design and production methods, adopting new processes and equipment, improving comprehensive utilization efficiency, and achieving energy savings and emission reductions; and green products refer to technical innovations that do not damage or that reduce damage to ecological environments during or after the use of the products [33,34]. Therefore, we define green innovation as a technological innovation that improves comprehensive utilization efficiency and achieves the purposes of saving energy and reducing emissions by enhancing the process, improving the design, and using alternative renewable energy. The green innovation of enterprises includes both green innovation input and green innovation output. However, since it is difficult to separate the green innovation input from enterprises’ R&D input, in this paper the number of green patents applied was used to measure the enterprise’s green innovation. The data on corporate green innovation were collected by the authors from the State Intellectual Property Office website; these included the green patent applications of the main industrial listed companies, wholly owned subsidiaries, holding subsidiaries, and joint ventures. The data on environmental taxes were derived from the notes to the financial statements in the annual reports of enterprises. The controlled variables were derived from the China Stock Market and Accounting Research Database (CSMARD). Furthermore, in order to prevent the effect of outliers, all continuous variables were Winsorized by 1% before and after.

### 3.5. Exit of Companies from Markets

This paper identified polluting enterprises via pollution-emission data. During the study, enterprises were cross-compared with the database of Chinese industrial enterprises to identify their existence and status [35]. To be exact, the data on Chinese industrial enterprises and pollution emissions from 1998–2014 were selected for cross-comparison. Firstly, we referred to Brandt (2012) to process the industrial-enterprise database and the pollution-emission database; secondly, we matched with the pollution-emission database according to the enterprise name and year and according to the unified social credit codes and year, merged the matching data in the second and third steps, and removed duplicate data; finally, we determined enterprises that satisfied the matching in the second or third step. By observing about 250 variables in the 16-year period, about 700,000 figures were obtained with an average matching rate of about 17% [36].

### 3.6. Other Variable Indicators

The control variables included the area of the administrative district, population, proportion of secondary industry, proportion of tertiary industry, proportion of investment, proportion of real-estate investment, proportion of foreign direct investment, distance to the nearest port, and number of patents. The data were derived from the statistical yearbook of the corresponding year of each city; in order to ensure the comparability of data between different years, the data were deflated according to the CPI of the current year. In order to prevent the impact of outliers, all continuous variables were shrunk by 1% before and after. Table 1 shows the results of descriptive statistics.

## 4. Pattern of Changes in Green Productivity through Time and in Different Regions

### 4.1. Changes in Green Productivity through Time and in Different Regions

In Table 2, the change in green productivity over the sample period was not significant, but the overall trend was that the national average green productivity increased after 2014 compared to the previous years. This change in green productivity may have been related to the faster development of the digital economy in the country after 2014. In addition, the changes in the standard-deviation indicators of the digital economy in different years were analyzed using the inter-regional digital economy standard-deviation indicators. It was found that the variance in green productivity across regions nationwide was 0.02806 in 2011 and 0.02192 in 2019, which revealed a gradual expansion.

This study visualized the spatial distribution of green productivity by region in the country from 2011 to 2019 using ArcGIS. The results are shown in Figure 1, in which the different colors represent different levels of green productivity. The deeper the color, the higher the GTFP value, thereby indicating higher levels of green productivity in the city. Figure 1 shows that most of the Yangtze River Delta and the Pearl River Delta regions had green productivity in the range of 1.00–1.06 (only a very small number of cities are below the left side of the range in a limited number of years), which also showed that the above-mentioned regions had high green productivity levels relative to other areas in China. The western region, on the other hand, faced both ecological and developmental pressures and had a relatively low level of green productivity. However, it is worth noting that in 2019, the green productivity of most cities in Sichuan and Guanzhong Plain, as well as Chongqing, was between 0.99 and 1.05, which indicated that the green productivity of some metropolitan areas in the western region showed an upward trend, thereby implying that green productivity will become an important factor in the coordinated development of China’s regional economy. In the provincial administrative regions, before 2015 there were still some provincial capitals with relatively low green productivity; however, after 2015 the green productivity of most provincial capitals was higher than 1.0, which indicated an obvious divergence in economic efficiency within the administrative regions. The subsequent analysis showed that this may have some correlation with the level of development of the regional digital economy.

### 4.2. Spatial Distribution of the Principal Components of the Digital Economy

This subsection investigates the spatial and temporal patterns of digital-economy development levels. These were based on the total digital-economy component index of each city in the country in this study, which were used to visualize the spatial distribution of the digital-economy development levels of each region in the country from 2011 to 2019 using ArcGIS tools (see Figure 2). In Figure 2, the different colors represent different levels of digital-economy development. The deeper the color, the larger the value of the digital economy’s principal component, indicating a higher level of digital-economy development in the city. Figure 2 shows that in 2011, the overall development level of China’s digital economy was relatively low, with most regions having a digital economy principal component below 90,000 and only a few cities such as Beijing and Shanghai over 90,000. In 2019, although most cities still had a digital-economy principal component index below 90,000, there was a significant increase in the number of cities over 90,000 compared to 2011; this increase was concentrated in the provincial capitals of each province. The reason for this phenomenon may be that China’s digital economy has taken a leading role in the development of large cities such as provincial capitals. This also would explain the changes in green productivity in Chinese cities shown in Figure 2.

### 4.3. Spatial Autocorrelation Analysis

To determine whether there was a spatial correlation in terms of the green productivity of China, a spatial-weights matrix based on geographic adjacency was built to calculate the Moran’s I of GTFP from 2011 to 2018. The results are shown in Table 3. Generally, the Moran’s I did not show a linear trend and was not significant under 10%. It can be assumed that this does not contradict H0 (the data were randomly distributed). This proved that the observed spatial model could be random. From 2011 to 2018, the variance in Moran’s I was as small as 0.0012. This deviation was relatively small. These results showed that urban green productivity was not affected by neighboring cities and did not show notable spatial clusters.

To determine whether there was a correlation in terms of the growth of the digital economy, a spatial-weights matrix based on geographic adjacency was built to calculate the Moran’s I of the main components of the digital economy from 2011 to 2018. The results are shown in Table 4. The Moran’s I was significant under 1%, thus refuting H0 (the data were randomly distributed). This proved that the observed spatial model was unlikely to be random. In addition, from 2011 to 2018, the variance in Moran’s I was as small as 0.0025. This deviation was rather small. These results showed that the degree of the digital economy was not randomly dispersed. On the contrary, the degree of the digital economy of a city was affected by neighboring regions. The Moran’s I was negative, meaning that the degree of the digital economy was not clustered but instead dispersed. The Moran’s I rose from −0.0590 in 2011 to 0.0359 in 2018, which demonstrated that the degree of the digital economy became less dispersed over time. These results corresponded to the facts. The degree of the digital economy in China was sporadic, unlike environmental pollution, which spread out from the center.

## 5. Empirical Results

### 5.1. Model Testing

The above results showed that there was a significant correlation between the digital economy and the regional green TFP. To further investigate the impact of the degree of digital-economy development on green total-factor productivity and the corresponding impact mechanism, this paper used the ordinary least squares (OLS) model. In this study, the green-productivity rates of 275 cities in China were used as the explained variables and the relevant indicators, which included the principal component of the digital economy, area of the administrative district, population, proportion of secondary industry, proportion of tertiary industry, proportion of investment, proportion of real-estate investment, proportion of local fiscal expenditure, proportion of local fiscal revenue, proportion of foreign direct investment, distance to the nearest port, and number of patents, were used as the explanatory variables. A regression analysis was applied using Stata 15.0 to explore the factors that may have affected green total-factor productivity. In order to prevent an estimation bias caused by the interaction of the indicators, a multicollinearity test was conducted on the above indicators. The results are shown in Table 5. The variance inflation factor (VIF) of each indicator was less than 10. Therefore, there was no multicollinearity relationship between the selected indicators.

### 5.2. Empirical Results

The estimation results of the regression model are given in Table 6 below. In the regression results in column (1), the coefficient of the core explanatory variable—the degree of digital-economy development—is positive and significant with a coefficient of 0.0041. Columns (2)–(6) add the control variables of the area of the administrative district, population, proportion of secondary industry, proportion of tertiary industry, proportion of investment, proportion of real-estate investment, proportion of local fiscal expenditure, proportion of local fiscal revenue, proportion of foreign direct investment, distance to the nearest port, and number of patents, respectively. As the controlled variables increased gradually, the coefficient of the digital-economy index remained positive and passed a 1% significance. Furthermore, the regression results were robust, which indicated that the digital economy increased the green total-factor productivity of cities. Therefore, H1 was verified. In the coefficient estimation results in column (6), we can see that for every 1% increase in the level of digital-economy development, the regional green total-factor productivity increased by 0.083%.

### 5.3. Instrument Estimate

The selection of appropriate instrumental variables as the core explanatory variables is the main approach to addressing the endogeneity problem. The method described by Huang Qunhui et al. (2019) uses the historical postal and telecommunications data for each city in 1984 as the instrumental variable for the composite index of digital economic development. However, since the number of cities in 1984 was relatively small, a large number of data are lost if the 1984 postal historical data are used. However, this paper drew on the method of Huang Qunhui et al. (2019). In this paper, the Internet-penetration rate of each prefecture-level city in 2001 was used as an instrumental variable for the development level of the digital economy in each region. On one hand, the development of the digital economy relies on the popularity of the Internet; regions with high penetration rates of Internet technology can nurture mature digital economies. The local historical Internet-penetration rate influences the development of the digital economy in subsequent stages through factors such as the technology level and usage habits. On the other hand, the digital economy is stripped down to the Internet industry, which is a higher-level industrial business that satisfies exclusivity. It should be noted that the original data of the selected instrumental variables were in cross-sectional form and could not be directly used in the econometric analysis of panel data. Referring to the solution by Nunn and Qian (2014) regarding this issue, a time-varying variable was introduced to construct the panel instrumental variable. Specifically, the number of national Internet users in the previous year was constructed as an interaction with the Internet-penetration rate of each city in 2001, respectively, as an instrumental variable for the city’s digital-economy index in that year.

Table 7 demonstrates the regression results of the instrumental variables. The coefficients of the degree of digital economy in columns (1) to (2) are all positive and significant at 1%, which showed that the effect of the digital economy on green-productivity enhancement still held after taking into account the endogeneity problem. In addition, to test the original hypothesis of “insufficient identification of instrumental variables”, the *p*-value of the LM statistic of the Kleibergen–Paap rk was 0.000, which meant that in the testing of the weak identification of the instrumental variables, the Wald F-statistic of Kleibergen–Paap rk was larger than the critical value of the Stock–Yogo weak identification test at 10%. This significantly rejected the original hypothesis. Therefore, there was no problem concerning weak instrumental variables. Therefore, the selection of the historical Internet-penetration rate of each city and the number of national Internet users in the same year as the instrumental variable for the level of digital-economy development was reasonable.

### 5.4. Regional Heterogeneity

The degree of economic development varies significantly in different regions of China, as does the degree of digital-economy development. The eastern region is relatively developed and therefore enjoys better Internet infrastructure and faster progress. Moreover, due to the differences in economic development across regions, the impact of the digital economy on green productivity also varies. In this section, cities in the eastern, central, western, and northeastern regions were compared and contrasted in terms of the impact of the digital economy on green productivity as shown in Table 8. The coefficients of the digital-economy degree in all the columns in Table 8 are notably positive, which proved that the digital economy increased regional green productivity. Column (1) shows that the degree of digital economy in the eastern region significantly improved the green productivity of this region due to its advanced digital-communication technology and infrastructure. The same presumably held for the northeastern region, for which the absolute value of the estimated coefficient was slightly larger than that of the eastern region, thereby showing a stronger influence of the digital economy on green productivity. The estimations for the central and western regions were similar; the absolute values of their estimated coefficients were smaller than that of the eastern region. The possible reasons for this were the later start to the building of Internet infrastructure, poor urbanization progress, and insufficient utilization of Internet infrastructure. Therefore, the positive effect of the digital economy on green productivity in the central and western regions was generally smaller than that in the eastern and northeastern regions.

### 5.5. Heterogeneity in Terms of Urban Scale

Compared with small cities, large cities boast the economic externality of urban agglomeration characterized by matching, sharing, and learning, which encourages resource sharing and enhances spillover effects. The advantages of cities in terms of expert knowledge and diversity, manpower, and information networks encourage technological innovation and application, which attracts digital industries. In addition, for a digital startup, in order to work more effectively as a medium for transactions, it is crucial to build a user base and improve services. This requires an Internet company to take active measures to consolidate and expand its user group and to form a mutually beneficial mechanism with other market players, thus realizing the goals of acquiring users and rapidly increasing installation. Therefore, the degree of digital economy varies in cities with different populations. This section aims to determine the effect of the digital economy on the green productivity of cities with different populations.

In this section of the study, cities were categorized as small, medium, and large according to the population benchmarks of 1 million and 5 million in order to study the effect of the digital economy on the green productivity of each category. Table 9 shows the regression results of each category. In Column (1), the regression coefficient of the digital economy is not significant, showing that green productivity was not sensitive to the degree of the digital economy. This is probably because the small cities had small populations that were not conducive to the growth of digital economies, or because the multiplier effect of the digital effect was not strong enough to produce a notable effect on green productivity. In contrast, the coefficients of the digital economies of the large and medium cities were significantly positive, which indicated that in cities with a certain population, the degree of the digital economy produced a notable effect on the increase in green productivity. The estimated coefficients in Columns (2) and (3) show that the absolute values of the coefficients of the large and medium cities were close and that when the urban population reached a certain benchmark, the elasticity of the effect of the digital economy on green productivity essentially remained the same.

### 5.6. Digital Economy and Spatial Distribution of Green Invention Patents

This section focuses on the spatial and temporal patterns of green invention patents. This study visualized the spatial distribution of green invention patents in each region of the country from 2011 to 2019 using ArcGIS based on the green patent invention indicators in each city of the country; specifically, the total number of green-invention patent applications and the total number of green-utility patent applications (see Figure 3). In Figure 3, the different colors represent different numbers of green patent applications. The deeper the color, the greater the total number of green patents applied for in that year in that city, thereby indicating a higher level of green innovation in the city. Figure 3 shows that from 2011 to 2019, the number of green-invention patent applications in China was relatively small, and the number of green-invention patent applications in a considerable number of cities was 0. The distribution of cities with useful green-invention patents was consistent with their level of digital economic development; it is not difficult to find that the cities with a higher green productivity were generally also the cities with higher numbers of green-invention-patent applications, which implied an inherent connection between the level of digital-economy development, green-invention patents, and green productivity.

This section of the study tested whether the impact of the digital economy as described in the previous section included an increase in the number of green patents held by companies. Using the database of listed companies from 2012 to 2019, this study examined the mechanism of the impact of the digital economy on local green productivity through a regression approach by using the data related to green-invention patents in the database. Table 10 reports the results of the analysis. In the regression results in column (1), the coefficient of the degree of digital-economy development is significantly positive. This indicated that digital-economy development helped to enhance the number of green patents of enterprises. Column (2) adds firm-level control variables for which the coefficient estimates and significance of the core explanatory-variable digital-economic development level in the regression results remained consistent. This indicated that the conclusions of this paper still held when the influencing factors at the enterprise level were controlled. When combining the regression coefficients in column (2), it is easy to see that the number of green-invention patents of enterprises increased by 128.9 for every 1% increase in the degree of digital-economy development. Columns (3)–(4) replace the explained variables with enterprises’ uses of green patents; the coefficient of the level of digital-economy development was still significantly positive, again indicating that digital-economy development helped to increase the number of green patents from enterprises. In terms of the coefficient value, the number of utility patents was comparatively small, which indicated that the number of green-utility patents needs to be increased.

### 5.7. Digital Economy and Exit of Polluting Enterprises from Markets

This section focuses on the spatial pattern of the exit from the market of polluting enterprises. Based on the results after matching the industrial-enterprise database and the pollution-emission database, this study presents the data concerning the market exit of polluting industrial enterprises in 2011 according to the accounting standards of enterprise survival. Furthermore, the spatial distribution of polluting industrial enterprises in each region of the country in 2011 was visualized by using ArcGIS (see Figure 4). In Figure 4a, the different colors represent different numbers of polluting enterprises. The deeper the color, the greater the number of emitters that exited the market in the city in that year, thereby indicating that the city eliminated more polluting enterprises. Combined with the results in Figure 1, it is easy to see that cities with a higher number of polluting enterprises exiting the market were generally also cities with higher green productivity; this conclusion still held when the indicator was replaced with the proportion of polluting enterprises exiting the market, which implied that there was a certain intrinsic connection between the level of digital economic development, polluting enterprises exiting the market, and green productivity.

Next, the question of whether the level of digital-economy development could accelerate the exit of polluting enterprises from the market was tested. In this paper, we used the industrial-enterprise database and enterprise-pollution data to match and determine the number of enterprises entering and exiting based on their operation. Since the only years in which the industrial-enterprise-database data and the years studied in this paper overlapped were 2011 and 2012, and considering that it was not possible to determine whether the enterprises exited the market in 2012, the industrial-enterprise data from 2011 were selected as the research sample for this section. Table 11 reports the results of the analysis. In the regression results in column (1) of Table 11, the coefficient of the level of digital-economy development is significantly positive. This indicated that the number of polluting enterprises exiting the market was higher in regions with a higher level of digital-economy development. The coefficient of the level of digital-economy development is significantly positive for column (2) in Table 7 when replacing the explained variable with the probability of enterprises exiting the market. This indicated that the development of the digital economy accelerated the exit of polluting enterprises. When combining the regression coefficients in column (2) of Table 8, it is easy to see that for every 1% increase in the level of digital-economy development, the probability of polluting enterprises exiting the market increased by 7.69%. This indicated that the development of the digital economy enhanced the green productivity of the region by encouraging the exit of polluting enterprises from the market.

## 6. Conclusions

Based on the fact that the digital economy has greatly influenced socioeconomic development, the green total-factor productivity and decomposition values of China from 2011 to 2019 were measured from the perspective of green development with the help of the DEA-GML index. In order to determine the development degree of the digital economy, the impact of the digital economy on green total-factor productivity and its mechanism was empirically tested in multiple dimensions using empirical regression. This study showed that the degree of the digital economy was conducive to the development of urban GTFP and that the urban GTFP was enhanced through two mechanisms: enhancing the production technology of enterprises and phasing pollution-intensive enterprises out of the market.

First, the role of the digital economy in increasing urban green development should be enhanced. Therefore, it is necessary to increase the government’s focus on the digital economy. The government should fully leverage the digital economy to increase the guidance of regional green development. It should cultivate diversified investment bodies; increase investment in all aspects of the digital economy; improve the construction of information and communication infrastructure; facilitate digital-technology research and development; encourage the popularization of practical applications such as artificial intelligence, big data, and the Internet of Things; transform high-energy-consuming and crude-production methods using modern networked and intelligent platforms; improve the allocation of factors in order to increase their combined efficiency; and continuously harness the positive effects of the digital economy on high-quality development.

Second, it is important to strengthen the penetration capacity of the digital economy and accelerate the deployment of the digital economy. Only after the digital economy exerts its scale effect can the urban GTFP be significantly enhanced. Therefore, it is necessary to deepen the integration of the digital economy and traditional industries, help traditional industries digitize and intellectualize, enrich digital knowledge and real digital-application scenarios, and increase the urban GTFP.

Third, it is important to encourage the advanced industrial structure and green development of cities. Research shows that the development of urban GTFP can be increased only when the industrial structure is advanced. The government should prioritize the development of advanced manufacturing industries, actively participate in the division of labor in the advanced global value chain, encourage green technological innovation, and gradually eliminate highly polluting enterprises with low capacity and high energy consumption while also focusing on supporting enterprises with key technologies in hand and strong demonstration effects as well as connecting upstream and downstream supply chains to drive the development of urban GTFP.

A fourth aim should be to strengthen the coordination and cooperation between the digital economy and various factors such as market-oriented reforms, talent training, and institutional governance to develop with greater synergy. It is important to actively integrate traditional industries; integrate digital technology innovation into all aspects of development, production, and application; create technological breakthroughs; and shift green total-factor productivity from a paradigm led by efficiency improvement to one led by technological progress in order to achieve the goal of the green development of the economy.

## Figures and Tables

**Figure 1 ijerph-20-01442-f001:**
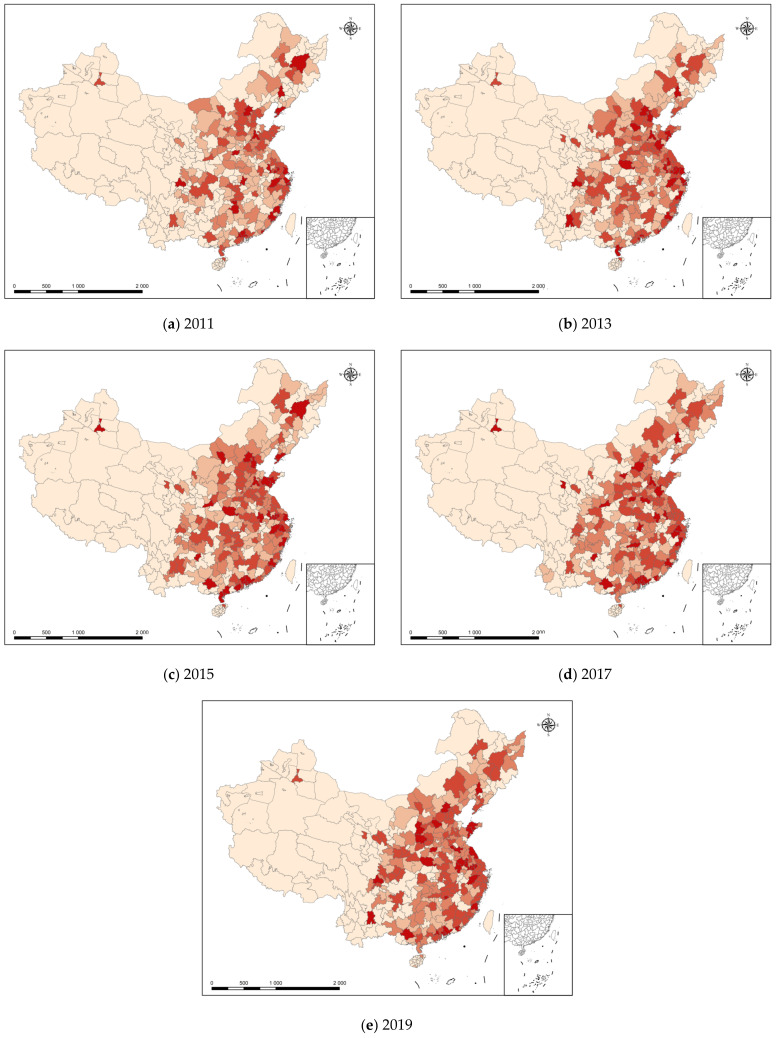
Spatial Distribution of the National Green Productivity Indexes (2011–2019).

**Figure 2 ijerph-20-01442-f002:**
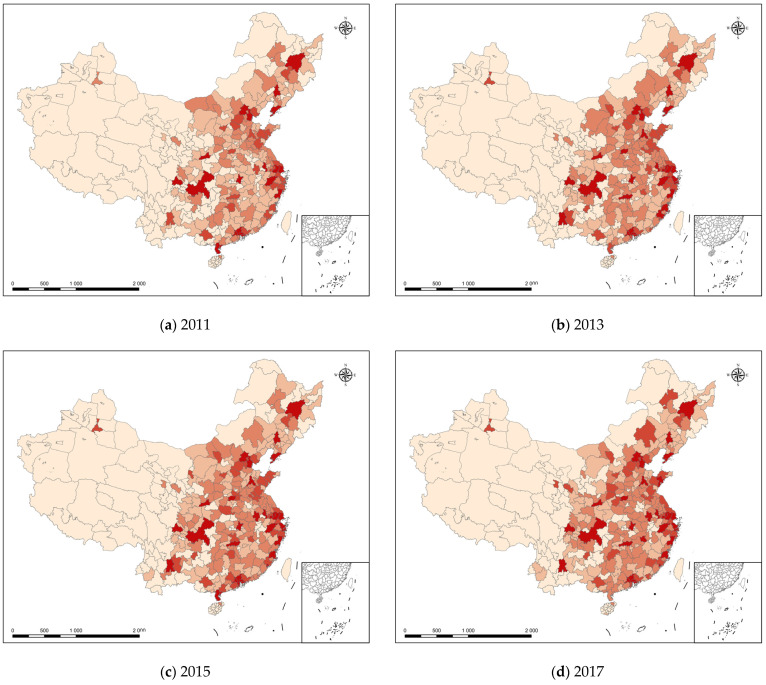
Spatial Distribution of the Principal Components of National Digital Economy (2011–2019).

**Figure 3 ijerph-20-01442-f003:**
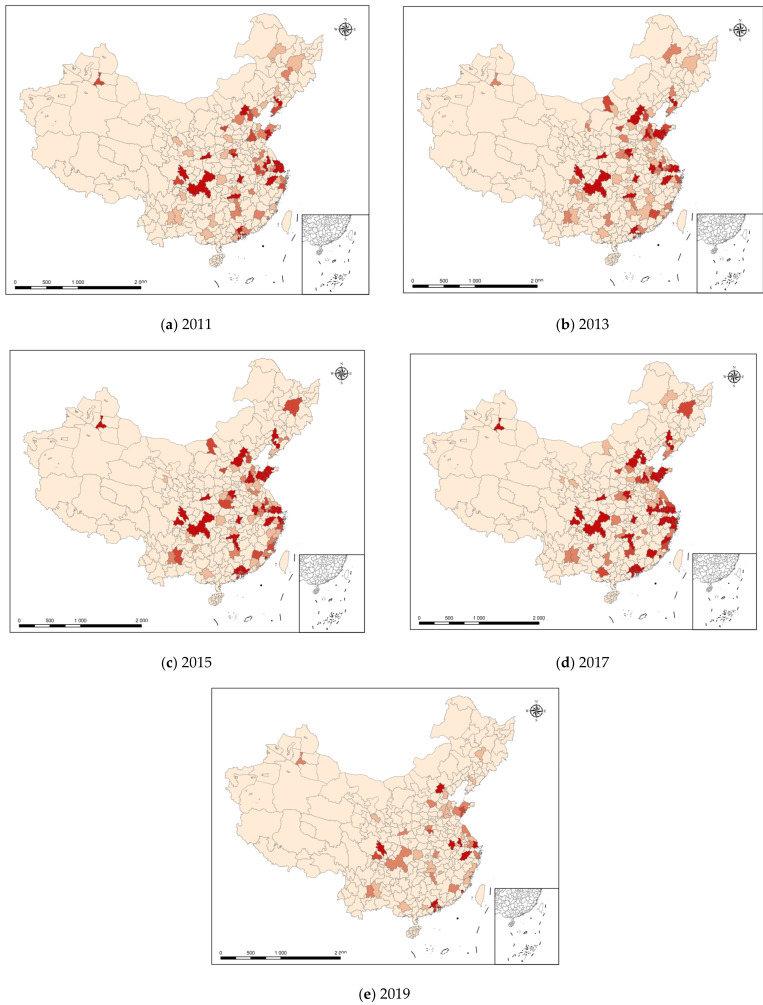
National Spatial Distribution of Green Invention Patents (2011–2019).

**Figure 4 ijerph-20-01442-f004:**
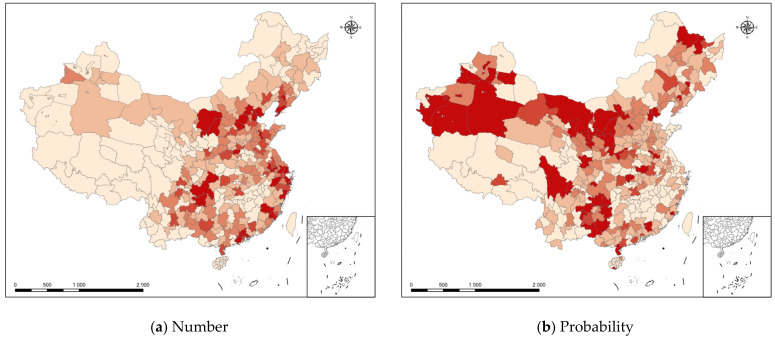
Spatial Distribution of Polluting Enterprises Exiting the Market.

**Table 1 ijerph-20-01442-t001:** Results of Descriptive Statistics.

Variable	Obersvations	Mean	Sd(I)	Min	Max
area	2475	0.002	0.002	0	0.02
pop	2475	0.006	0.001	0.003	0.007
sec	2475	0.048	0.011	0.012	0.089
thi	2475	0.04	0.01	0.01	0.079
fdi	2475	0.078	0.147	0	1.4
to_port	2475	0.471	0.423	0.002	2.762
inv	2475	0.016	0.001	0.013	0.018
est	2475	0.003	0.004	0	0.034
pat	2475	0.006	0.012	0	0.167

**Table 2 ijerph-20-01442-t002:** Trend of Green Productivity Indices in Different Regions across China (2011–2019).

2011	2012	2013	2014	2015	2016	2017	2018	2019
Average								
0.9650	1.0012	0.9615	0.9867	1.0032	1.0004	0.9854	1.0444	1.0403
Standard Deviation								
0.0280	0.0291	0.0290	0.0292	0.0278	0.0294	0.0289	0.0286	0.0292

**Table 3 ijerph-20-01442-t003:** Moran’s I of the GTFP (2011–2018).

Year	Moran’s I	Expectation Index	Sd(I)	Z Score	*p* Value
2011	−0.0049	−0.0041	0.0026	−0.2984	0.7654
2012	−0.0059	−0.0041	0.0026	−0.6843	0.4938
2013	−0.0022	−0.0041	0.0026	0.7468	0.4552
2014	−0.0027	−0.0041	0.0026	0.5744	0.5657
2015	−0.0049	−0.0041	0.0026	−0.2948	0.7681
2016	−0.0016	−0.0041	0.0026	1.0073	0.3138
2017	−0.0040	−0.0041	0.0026	0.0685	0.9454
2018	−0.0015	−0.0041	0.0026	1.0422	0.2973

**Table 4 ijerph-20-01442-t004:** Moran’s I of the Degree of Digital-Economy Development (2011–2018).

Year	Moran’s I	Expectation Index	Sd(I)	Z Score	*p* Value
2011	−0.059	−0.0041	0.0025	−21.5451	0.0000
2012	−0.0463	−0.0041	0.0025	−16.5583	0.0000
2013	−0.0484	−0.0041	0.0025	−17.394	0.0000
2014	−0.0417	−0.0041	0.0025	−14.7345	0.0000
2015	−0.0397	−0.0041	0.0025	−13.9514	0.0000
2016	−0.0384	−0.0041	0.0025	−13.4697	0.0000
2017	−0.0384	−0.0041	0.0025	−13.4656	0.0000
2018	−0.0359	−0.0041	0.0025	−12.4967	0.0000

**Table 5 ijerph-20-01442-t005:** Descriptive Statistics of the Explanatory Variables.

Variable	VIF	1/VIF
LnDigital	5.98	0.167143
area	1.95	0.511755
lnpop	4.49	0.222508
Thi	6.58	0.151973
Sec	5.34	0.187404
Lnp	3.11	0.321844
fdi	3.29	0.304367
to_port	16.27	0.061459
inv	6.12	0.163383
est	4.95	0.202026
pat	3.74	0.267388
Mean VIF	3.46	

**Table 6 ijerph-20-01442-t006:** Influence of Digital-Economy Degree on Regional Green Productivity.

	(1)	(2)	(3)	(4)	(5)	(6)
LnDigital	0.075 ***	0.068 ***	0.068 ***	0.076 ***	0.079 ***	0.083 ***
	(79.10)	(55.65)	(40.59)	(39.63)	(40.04)	(41.15)
area		−0.882	−0.458	−0.757	−0.736	−0.728
		(−1.62)	(−0.79)	(−1.29)	(−1.19)	(−1.18)
pop		13.440 ***	15.197 ***	13.497 ***	11.942 ***	9.378 ***
		(7.88)	(7.40)	(6.04)	(5.23)	(4.06)
sec			0.368 **	0.572 ***	0.531 ***	0.603 ***
			(2.46)	(3.39)	(3.12)	(3.48)
thi			0.081	0.584 ***	0.497 **	0.526 **
			(0.43)	(2.86)	(2.43)	(2.55)
inv				2.697	4.847 **	3.227
				(1.29)	(2.29)	(1.54)
est				−2.819 ***	−2.280 ***	−1.218 ***
				(−10.21)	(−6.85)	(−3.43)
fdi					−0.037 ***	−0.022 **
					(−4.31)	(−2.54)
to_port					13.219 *	11.431 *
					(1.88)	(1.65)
pat						−0.902 ***
						(−7.94)
Province	Yes	Yes	Yes	Yes	Yes	Yes
Year	Yes	Yes	Yes	Yes	Yes	Yes
cons	0.503 ***	0.473 ***	0.446 ***	0.324 ***	0.275 ***	0.276 ***
	(54.88)	(48.24)	(28.55)	(13.18)	(10.66)	(10.80)
N	2475	2475	2475	2475	2475	2475
R^2^	0.791	0.796	0.798	0.807	0.810	0.815
adj. R^2^	0.788	0.793	0.794	0.804	0.806	0.811

Note: *t* statistics in parentheses; * *p* < 0.1, ** *p* < 0.05, *** *p* < 0.01.

**Table 7 ijerph-20-01442-t007:** Regression of Instrumental Variables.

	(1)	(2)
LnDigital	0.0744 ***	0.116 ***
	(54.18)	(9.95)
area		−1.943 *
		(−1.81)
pop		−6.193
		(−0.96)
sec		0.0932
		(0.39)
thi		−0.599
		(−1.49)
inv		−5.762
		(−1.41)
est		−1.925 ***
		(−4.46)
fdi		−0.0258 ***
		(−2.81)
to_port		7.141
		(0.92)
pat		−1.386 ***
		(−6.78)
_cons	0.508 ***	0.310 ***
	(39.81)	(10.06)
N	2475	2475
R^2^	0.785	0.791

* *p* < 0.1, *** *p* < 0.01.

**Table 8 ijerph-20-01442-t008:** Influence of Degree of Digital Economy on Green Productivity: Regional Heterogeneity.

	(1)	(2)	(3)	(4)
	East	Central	Northeast	West
LnDigital	0.0865 ***	0.0851 ***	0.0913 ***	0.0786 ***
	(20.08)	(22.77)	(9.95)	(20.25)
area	−0.582	−2.327 *	−5.525	−0.906
	(−0.22)	(−1.71)	(−1.35)	(−0.98)
pop	12.07 ***	10.80 **	19.63 *	1.961
	(3.11)	(1.97)	(1.93)	(0.40)
sec	1.967 ***	−0.0611	1.052 ***	0.555
	(4.29)	(−0.14)	(2.69)	(1.52)
thi	1.277 **	−0.0626	1.336 **	0.919 **
	(2.46)	(−0.13)	(2.52)	(2.07)
inv	11.42 **	4.147	−7.911 *	8.596 *
	(2.49)	(0.89)	(−1.69)	(1.90)
est	−1.991 ***	−1.024	1.218	0.295
	(−3.55)	(−1.10)	(0.84)	(0.28)
fdi	−0.00256	−0.0149	−0.0176	−0.0265
	(−0.17)	(−0.70)	(−1.03)	(−0.64)
to_port	1.013	14.34	30.93	22.53
	(0.05)	(1.36)	(1.12)	(1.55)
pat	−0.946 ***	−0.688	−4.403 ***	−2.422 **
	(−6.32)	(−0.88)	(−2.85)	(−2.30)
Province	Yes	Yes	Yes	Yes
Year	Yes	Yes	Yes	Yes
cons	0.0885	0.283 ***	0.251 ***	0.255 ***
	(1.40)	(5.01)	(3.62)	(4.67)
N	783	810	306	576
R^2^	0.741	0.787	0.827	0.805
adj. R^2^	0.731	0.779	0.813	0.795

* *p* < 0.1, ** *p* < 0.05, *** *p* < 0.01.

**Table 9 ijerph-20-01442-t009:** Influence of Degree of Digital Economy on Green Productivity: Heterogeneity in Terms of Urban Scale.

	(1)	(2)	(3)
LnDigital	0.0321	0.0865 ***	0.0759 ***
	(1.61)	(34.17)	(20.07)
area	14.21	−0.243	2.066
	(0.47)	(−0.29)	(0.73)
pop	14.28	6.546 *	9.466
	(0.12)	(1.83)	(1.37)
sec	−3.890	0.527 **	0.316
	(−0.92)	(2.36)	(0.81)
thi	−3.397	0.386	0.0223
	(−0.71)	(1.47)	(0.05)
inv	−1.633	2.402	6.456
	(−0.07)	(0.88)	(1.48)
est	4.316	1.457 *	−1.258 ***
	(0.32)	(1.81)	(−2.71)
fdi	−1.269	−0.0146	−0.0206 **
	(−1.58)	(−0.65)	(−2.12)
to_port	−507.8	11.97	3.984
	(−0.87)	(1.24)	(0.32)
pat	64.78	−1.264 ***	−0.724 ***
	(1.03)	(−6.88)	(−4.37)
Province	Yes	Yes	Yes
Year	Yes	Yes	Yes
_cons	1.613 *	0.276 ***	0.316 ***
	(1.76)	(7.65)	(4.86)
N	85	1542	848
R^2^	0.652	0.789	0.716
adj. R^2^	0.455	0.782	0.702

* *p* < 0.1, ** *p* < 0.05, *** *p* < 0.01.

**Table 10 ijerph-20-01442-t010:** Effect of Digital Economy on Corporate Applications for Green Patents.

	(1)	(2)	(3)	(4)
LnDigital	488.5 ***	128.9 ***	588.4 ***	137.3 ***
	(33.64)	(6.03)	(33.61)	(5.39)
area		14.48 **		18.95 **
		(2.15)		(2.36)
pop		−107.4 ***		−135.7 ***
		(−4.33)		(−4.59)
sec		0.533		−2.817
		(0.29)		(−1.28)
thi		−7.759 ***		−7.196 ***
		(−3.49)		(−2.72)
inv		−159.6 ***		−168.8 ***
		(−6.95)		(−6.17)
est		121.4 ***		128.6 ***
		(33.62)		(29.87)
fdi		423.4 ***		924.5 ***
		(4.55)		(8.33)
to_port		−97.77		−180.4 **
		(−1.28)		(−1.98)
Province	Yes	Yes	Yes	Yes
Year	Yes	Yes	Yes	Yes
cons	−419.667 ***	213.503 ***	−518.688 ***	240.309 ***
	(−29.80)	(7.63)	(−30.56)	(7.21)
N	2475	2475	2475	2475
R^2^	0.464	0.736	0.426	0.711
adj. R^2^	0.456	0.730	0.418	0.706

** *p* < 0.05, *** *p* < 0.01.

**Table 11 ijerph-20-01442-t011:** Influence of Digital Economy on the Operational Status of Polluting Enterprises.

	(1)	(2)	(3)	(4)
LnDigital	10.16 ***	5.536 **	0.0822 ***	0.0769 ***
	(9.22)	(2.17)	(6.99)	(6.99)
area		838.6		3.975
		(1.34)		(1.47)
lnpop		1977.2		21.63 **
		(0.81)		(2.04)
sec		359.2 **		1.317 *
		(2.10)		(1.79)
thi		189.5		1.128
		(0.85)		(1.17)
lninv		1615.5		14.36
		(0.55)		(1.14)
est		−483.9		−4.907 *
		(−0.73)		(−1.72)
fdi		35.91 ***		−0.0384
		(3.10)		(−0.77)
to_port		8103.1		46.61
		(1.11)		(1.48)
Province	Yes	Yes	Yes	Yes
_cons	−63.00 ***	−89.74 ***	−0.428 ***	−0.863 ***
	(−6.05)	(−2.61)	(−0.00)	(−5.81)
N	264	264	264	264
R^2^	0.476	0.529	0.656	0.708

* *p* < 0.1, ** *p* < 0.05, *** *p* < 0.01.

## Data Availability

Due to the confidentiality and privacy of the data, they will only be provided upon reasonable request.

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
