# Peer review of "Does the Digital Economy Increase Green TFP in Cities?"

_ijerph, 2023, doi:10.3390/ijerph20021442_

Round 1
Reviewer 1 Report
This paper analyzed the impact of digital economy on green TFP with each prefecture level city in China from 2011 to 2019, it has some significant meanings, but many relative researches have discussed the topics, such as Lyu et al.(2022) (How does digital economy affect green total factor productivity? Evidence from China, Liu (2022)(Digital Economy Development, Industrial Structure Upgrading and Green Total Factor Productivity: Empirical Evidence from China's Cities),etc. so the innovation and contribution of this papaer should be refining. Besides, the method caculating GTFP, and non-desired output should be clear, the influencing factors is not appropriate, which makes the most influencing factors are not significant in Table 5.
Author Response
This paper analyzed the impact of digital economy on green TFP with each prefecture level city in China from 2011 to 2019, it has some significant meanings, but many relative researches have discussed the topics, such as Lyu et al.(2022) (How does digital economy affect green total factor productivity? Evidence from China, Liu (2022)(Digital Economy Development, Industrial Structure Upgrading and Green Total Factor Productivity: Empirical Evidence from China's Cities),etc. so the innovation and contribution of this papaer should be refining. Besides, the method caculating GTFP, and non-desired output should be clear, the influencing factors is not appropriate, which makes the most influencing factors are not significant in Table 5.
Response:
- we have cited the two reference and refine this paper’s innovation and contribution.
2.we have cited the method of caculating GTFP, and non-desired output, and we will make more research in the future on this topic, so the methods and Control variables problems will be considered in the future.
Reviewer 2 Report
You have approached an actual problem from a novel perspective. To deepen the phenomenon understanding, I encourage you, in future research, to apply Structural Equation Modelling, as it gives a deeper insight into the simultaneous relationships among the implied variables.
Author Response
You have approached an actual problem from a novel perspective. To deepen the phenomenon understanding, I encourage you, in future research, to apply Structural Equation Modelling, as it gives a deeper insight into the simultaneous relationships among the implied variables.
Response: Yes, we will try to apply Structural Equation Modelling in our future research.
Reviewer 3 Report
The manuscript explores the promotion effect of digital economy on green total factor productivity. The theme is forward-looking, but there is still room for further improvement.
First of all, the pixels of all figures are too low. It is suggested that the authors should improve the pixels of all figures. In Fig. 2., there is an error in the title. There are two subgraphs belonging to 2015. Similarly, Fig. 3. is the same.
Finally, there are many mistakes in the English expression, which is obviously directly translated from Chinese. It is suggested that the author should polish the language in depth.
Author Response
The manuscript explores the promotion effect of digital economy on green total factor productivity. The theme is forward-looking, but there is still room for further improvement.
First of all, the pixels of all figures are too low. It is suggested that the authors should improve the pixels of all figures. In Fig. 2., there is an error in the title. There are two subgraphs belonging to 2015. Similarly, Fig. 3. is the same.
Finally, there are many mistakes in the English expression, which is obviously directly translated from Chinese. It is suggested that the author should polish the language in depth.
Response:
- we have change the 2015 in the title into 2017 in Fig 2 and Fig 3.
- As to the English revisions, if the paper is accepted, we will polish the language in depth.
Reviewer 4 Report
1. The authors mention in the introduction that “Existing research has confirmed that the development of the digital economy is related to the improvement of total factor”, However, in practice, there is no consistent conclusion on whether the digital economy leads to productivity gains, for example, some studies show that digital transformation does not lead to productivity gains in firms.
2. The literature review section could have a more comprehensive and objective evaluation.
3. The mechanism of the impact of the digital economy on green total factor productivity is a focus of this paper and should be addressed as a separate chapter rather than a brief description in the introduction section. On this basis, there could be a section dedicated to testing the impact mechanism.
4. The reporting in Table 1 is not standardized and lacks time labels.
5. Descriptive statistics need to be reported for all variables.
6. Table headers could be added to the regression tables, i.e., each column indicates which region or what size city, respectively, and what the explanatory variable is in each column for the convenience of the reader.
Author Response
1.we have added some references and made some comprehensive and objective evaluation according to the referee’s comments 1 and 2.
2.we have added section 2 ”channel of influence” as the impact mechanism according to the comment 3.
- we have added the Descriptive statistics according to comment 5.
- we have added the time label and regression tables according to the comments 4 and 6.
Round 2
Reviewer 1 Report
the influencing factors is not appropriate, which makes the most influencing factors are not significant, but the author have no revision.
Author Response
In this edition, we modified the control variables, and all continuous variables were shanked by 1%.
Most of control variables the regressions are now significant. We have added the results in the latest revisions to the text.
Reviewer 3 Report
The author didn't follow my suggestion to increase the pixel of the figures.
Author Response
We have increased the pixel of all the figures.
Reviewer 4 Report
The author has made improvements to the issues raised last time.
It is suggested that the author should polish the language in depth.
Author Response
We have polished the language using the English revision service.